# *Weizmannia coagulans* BC2000 Plus Ellagic Acid Inhibits High-Fat-Induced Insulin Resistance by Remodeling the Gut Microbiota and Activating the Hepatic Autophagy Pathway in Mice

**DOI:** 10.3390/nu14194206

**Published:** 2022-10-09

**Authors:** Long Jin, Hongyang Dang, Jinyong Wu, Lixia Yuan, Xiangsong Chen, Jianming Yao

**Affiliations:** 1Institute of Plasma Physics, Hefei Institutes of Physical Science, Chinese Academy of Sciences, Hefei 230031, China; 2University of Science and Technology of China, Hefei 230026, China; 3Probiotics Institute, Hefei 230031, China; 4College Life Science & Technology, Xinjiang University, Urumqi 830046, China; 5Institute of Nutrition and Health, Qingdao University, Qingdao 266021, China

**Keywords:** *Weizmannia coagulans*, ellagic acid, high-fat diet, insulin resistance, gut microbiota, autophagy

## Abstract

(1) Background: Ellagic acid (EA) acts as a product of gut microbiota transformation to prevent insulin resistance, which is limited by high-fat diet (HFD)-induced dysbiosis. The aim of this study was to investigate the synergistic effects and mechanisms of supplementation with the probiotic *Weizmannia coagulans* (*W. coagulans)* on the prevention of insulin resistance by EA; (2) Methods: C57BL/6J mice were divided into five groups (n = 10/group): low-fat-diet group, high-fat-diet group, EA intervention group, EA + *W. coagulans* BC77 group, and EA + *W. coagulans* BC2000 group; (3) Result: *W. coagulans* BC2000 showed a synergistic effect on EA’s lowering insulin resistance index and inhibiting high-fat diet-induced endotoxemia. The combined effect of BC2000 and EA activated the autophagy pathway in the mouse liver, a urolithin-like effect. This was associated with altered β-diversity of gut microbiota and increased *Eggerthellaceae*, a potential EA-converting family. Ellagic acid treatment alone and the combined use of ellagic acid and *W. coagulans* BC77 failed to activate the hepatic autophagy pathway; (4) Conclusions: *W. coagulans* BC2000 can assist EA in its role of preventing insulin resistance. This study provides a basis for the development of EA-rich functional food supplemented with *W. coagulans* BC2000.

## 1. Introduction

Ellagic acid (EA) is a phytochemical abundant in raspberries, blackberries, walnuts, and cranberries, among others [1]. The polarity of EA is very large, which is not conducive to its absorption in the intestinal tract, but its absorption efficiency is improved after being converted into urolithin A by the intestinal microbiota. Urolithin A has been suggested to be responsible for preventing insulin resistance by activating autophagy in pancreatic β cells [2] and the liver [3].

Recent investigations have shown large inter-individual differences in the biotransformation of EA because of the specific gut microbiome of each individual [4]. In individuals with metabolic syndrome, the gut microbiota converts EA to isourolithin A and/or urolithin B, which are less effective in metabolic regulation [5]. The study also showed that the gut microbiota of 5–25% of individuals was not capable of transforming EA at all. Some researchers expect to isolate EA-converting bacteria from the gut as probiotics to promote the biomodulatory effects of EA. Some strict anaerobic bacteria in the gut are involved in the conversion of EA to urolithin B. *Gordonibacter urolithinfaciens* was originally discovered as gut bacteria that could convert EA into urolithins [6]. Recently, it was found that some *Bifidobacterium pseudocatenulatum* [7] and *Ellagibacter isourolithinifaciens* [8] strains can also convert EA to urolithin A and B. However, these bacteria are not typical strains that can be used in food, especially according to Chinese regulations. Therefore, on the one hand, further research on edible probiotics that can transform EA is required. On the other hand, it is more necessary to investigate nutritional strategies to improve the efficiency of EA transformation by modulating gut microbiota.

One potential approach is to consume a combination of EA and probiotics that modulate the gut microbiota. However, traditional probiotics such as *Lactobacillus* and *Bifidobacterium* are very sensitive to temperature [9]. They cannot be added to nut foods that are kept at room temperature. It is therefore necessary to find a heat-resistant probiotic. *Weizmannia coagulans* (*W. coagulans*) is of increasing interest in different types of food due to its resistance to heat, stomach acid, and bile salt [10]. It can promote the maintenance of an anaerobic environment in the intestine, which is conducive to the proliferation of strictly anaerobic bacteria including EA-transformable bacteria. *W. coagulans* can affect the intestinal microbiota through temporary proliferation in the human intestine [11]. *W. coagulans* has been demonstrated to improve gut nutrient absorption and availability to facilitate digestion in conjunction with gut microbes [12]. *W. coagulans* was included by the National Health and Family Planning Commission of the People’s Republic of China in the “List of Usable Bacteria in Food” in 2016.

Therefore, the addition of *W. coagulans* may enhance the physiological regulation by EA. The present study was conducted to investigate the effects of two *W. coagulans* strains, BC2000 and BC77, on EA’s regulation of high-fat diet-induced insulin resistance. The findings of this study provide insight into the use of *W. coagulans* as a probiotic in EA-rich foods.

## 2. Materials and Methods

### 2.1. Materials

*W. coagulans* BC2000 and BC77 were provided by the laboratory of Chacha Food Co., Ltd. (Hefei, China), and were cultivated in MRS broth at 45 °C. EA was obtained from Shanghai Yuanye Biotechnology Co., Ltd. (Shanghai, China). Insulin, tumor necrosis factor α (TNF-α), interleukin 6 (IL-6), lipopolysaccharide (LPS), zonulin, and high-sensitivity C-reactive protein (hs-CRP) ELISA kits were purchased from Jiangsu Jingmei Biotechnology Co., Ltd. (Yancheng, China).

### 2.2. Evaluation of the Probiotic Properties of W. coagulans 

*W. coagulans* BC2000 (No. # 1) was evaluated against five commercially available species of *W. coagulans* (No. # 2–6) for heat, acid, and bile salt resistance. The test methods are described in Appendix A.

### 2.3. Experimental Design and Animals

A total of 50 C57BL/6J male mice (6 weeks old, 21 ± 0.8 g) were purchased from SPF Biotechnology Co., Ltd. (Beijing, China). The mice were reared in the Laboratory Animal Center of Qingdao University. The rearing environment is as follows: relative temperature: 22 °C ± 1 °C, relative humidity: 50% ± 5%, and light/dark cycle: 12 h/12 h. Mice had free access to food and water during rearing. Mice in each group were reared in 3 cages (3–4 mice/cage). The body weight and food intake of mice were measured and recorded weekly. Animal experimentation procedures were approved by Qingdao University Animal Ethics Committee (No. QDU-AEC-2022259) and were carried out based on according to the National Guidelines for Experimental Animal Welfare. 

After one week of acclimation, 50 mice were randomized to 5 groups (n = 10/group) to receive LFD, HFD (providing 60% of fat energy, Table 1), HFD + EA, HFD + EA + BC77, and HFD + EA + BC2000, respectively. EA was dosed at 0.3 g/kg of high-fat feed. The feed contained 0.1 g of lyophilized BC2000 or BC77 powder in 1000 g of feed. The probiotic powder content was 4 × 10^11^ CFU/g. Feed intake based on mice was 2.3–3.3 g/d. BC2000 or BC77 intake was 9.2 × 10^7^–1.32 × 10^8^ CFU/day/animal. Rearing lasted for 10 weeks. The body mass and food intake of mice were measured weekly.

### 2.4. Tissue Sample Collection

The mouse was anaesthetized, and blood was collected by removing the eyeball, which was then sacrificed by cervical dislocation. Blood was centrifuged at 4 °C and 1500 g for 10 min to collect the upper plasma. The plasma was aliquoted into centrifuge tubes and stored at −80 °C for later use. Each mouse was quickly dissected to collect its liver. The liver was added to normal saline and homogenized to prepare 10% (W/V) tissue homogenate. The homogenate was stored at −80 °C for subsequent procedures. The cecum contents were collected and stored in sterilized EP tubes at −80 °C for high-throughput sequencing of 16S rDNA.

### 2.5. Determination of Insulin Resistance Indicators

An oral glucose tolerance test was performed on mice one week before the completion of the study. Mice were gavaged with 400 mg/mL of glucose (5 mL per kg body weight) after fasting without water at night. Blood was collected from mouse tail veins at 0, 15, 30, 60, and 90 min after gavage. The collected blood was used to determine the blood sugar levels of the mice. At the end of the study, mice were fasted for 6 h and underwent blood sugar level measurement before anesthesia before sacrifice. Blood samples were collected and centrifuged at 4 °C and 1500 g for 10 min. Insulin levels were determined in strict accordance with the Instructions for Use of the ELISA kit (Jiangsu Jingmei Biotechnology Co., Ltd.). The plasma insulin levels were calculated based on the standard curve. The homeostasis model assessment-estimated insulin resistance (HOMA-IR) was computed using the following formula [13]:$$\mathrm{HOMA}-\mathrm{IR}=\frac{\mathrm{Fasting}\;\mathrm{blood}\;\mathrm{sugar}\;\mathrm{level}\;\left(\mathrm{mg}/\mathrm{dL}\right)\times \;\mathrm{Insulin}\;\mathrm{level}\;\left(\mathrm{mU}/L\right)}{22.5}$$

### 2.6. Determination of Inflammatory Cytokine Indicators

The plasma levels of TNF-α, IL-6, LPS, zonulin, hs-CRP and liver levels of TNF-α were determined strictly according to the Instructions for Use of the ELISA kit (Jiangsu Jingmei Biotechnology Co., Ltd.).

### 2.7. High-Throughput Sequencing of 16S rDNA in Colonic Contents

Total DNA in the cecum contents was extracted in strict accordance with the Instructions for Use of the DNA extraction kit. The DNA concentration was determined. The DNA was used as a template for PCR. Hypervariable regions V3 and V4 on prokaryotic 16S rDNA were selected to generate amplicons. V3 and V4 regions were amplified using the forward primer CCTACGGRRBGCASCAGKVRVGAAT and reverse primer GGACTACNVGGGTWTCTAATCC. The PCR amplification procedures were as follows: pre-denaturation at 94 °C for 3 min, denaturation at 94 °C for 5 s, annealing at 57 °C for 90 s, extension at 72 °C for 10 s, and final extension at 72 °C for 5 min, 24 cycles in total. The quality of PCR products was inspected through agarose gel electrophoresis (1.5% [*w*/*v*] agarose). The PCR-amplified library underwent PE250/PE300 paired-end sequencing in strict accordance with the Instructions for Use of the Illumina MiSeq System (Illumina, San Diego, CA, USA).

Sequencing data were analyzed with EasyAmplicon [14]. Species-level annotation was achieved using the Bayesian lowest common ancestor (BLCA) algorithm [15]. Data were filtered based on low counts (minimum count: less than 4, sample prevalence: less than 20%) and low variances (based on 10% of the interquartile range). Data were not scarce. Analysis of similarities (ANOSIM) based on the Bray–Curtis distance matrix between samples was used to perform constrained principal coordinate analysis (CPCoA) to assess inter-group differences. The family-level and genus-level analyses of mouse cecum contents were conducted using linear discriminant analysis (LDA) combined with effect size measurements (LEfSe). Bacteria with an LDA score > 2.00 and a *p* value < 0.05 were considered dominant in the group. The genus-level differences of microbiota were analyzed by the edgeR-based Wald test for pairwise comparison between groups.

### 2.8. Mouse Liver RNA Sequencing and Gene Enrichment Analysis

For transcriptome analysis, total RNA was extracted from four liver samples from each group using TRIzol reagent (Invitrogen, Carlsbad, CA, USA), examined by 1% agarose gel electrophoresis, and determined by NanoDrop™ Lite spectrophotometer (Thermo Fisher Scientific, Waltham, MA, USA) to determine its concentration and purity. RNA-seq transcriptome libraries were prepared with 1 μg of total RNA using the TruSeq RNA Sample Preparation Kit (Illumina, San Diego, CA, USA). Libraries were sequenced on the Illumina HiSeq 4000 platform (Illumina, San Diego, CA, USA). The R package “limma” (http://www.bioconductor.org/packages/release/bioc/html/limma.html) was used to normalize the RNA expression profiles. Differentially expressed genes (DEGs) were tested using Bayesian-adjusted t-statistics of limma’s linear model. A multiple testing correction based on false discovery rate (FDR) was performed. Log (fold change) (log2FC) >1.5 and *p* < 0.05 were used as cut-off criteria for the differentially expressed genes (DEGs) sample. Genomic enrichment analysis of genomes meeting this criterion was performed using the R package “Clusterprofiler v4.0” (http://www.bioconductor.org/packages/release/bioc/html/clusterProfiler.html) for biological process themes of the Gene Ontology (GO) and Kyoto Encyclopedia of Genes and Genomes (KEGG) pathways [16]. A corrected *p* < 0.05 was the cutoff criterion. KEGG pathways are visualized with Pathview.

### 2.9. Statistical Analysis

IBM SPSS Statistics 20.0 (SPSS, Chicago, IL, USA) was used to analyze the study data. The study results were expressed as mean ± standard deviation (SD). Data were plotted using GraphPad Prism 8.0 (GraphPad Software, Chicago, IL, USA). When variances in the study results were homogeneous, Tukey’s test based on one-way analysis of variance (ANOVA) was used for multiple comparisons. When the variances were not homogeneous, the Kruskal–Wallis test was used. *p* < 0.05 indicated a statistically significant difference.

## 3. Results

### 3.1. Probiotic Properties of W. coagulans

The viability of the 6 *W. coagulans* spores tested ranged from 76.86% to 94.66% after holding in a water bath at 80 °C for 30 min, with BC2000 and #5 being the highest, at 94.66% and 94.86%, respectively. After a 30 min water bath at 90 °C, the highest survival rate of spores was 77.58% for BC2000 (Appendix A). After holding at pH 2.0 and 37 °C for 2 h, the survival of the 6 tested *W. coagulans* spores ranged from 4.13% to 66.31%, with the highest survival rate of 66.31% for BC2000 (Appendix A). After 24 h of storage in a 0.03% sodium cholate solution, the survival of the 6 *W. coagulans* spores tested ranged from 83.28% to 100%, with BC2000 and #6 being the highest with 100% survival and #3 being the lowest at 83.28%, while the remaining samples were all greater than 90% viable. However, after 24 h of storage in 0.3% sodium cholate solution, samples BC2000 and #6 were the most tolerant to bile salts, with germination rates above 90% after treatment (Appendix A). These results indicate that *W. coagulans* BC2000 is resistant to heat, stomach acid, and bile salt.

### 3.2. Impact of W. coagulans and EA on Body Weight of High-Fat Diet-Fed Mice

As shown in Figure 1A, after five weeks of HFD feeding, the mouse body weight in the HFD group increased significantly compared with that in the LFD group (*p* < 0.05). At the end of the 10-week feeding, the mouse body weight in the HFD group was 1.2 times that in the LFD group. After 9 weeks of feeding, the mouse body weight in the HFD + EA group declined significantly compared with that in the HFD group (*p* < 0.05). However, mice in the HFD + EA + BC2000 group did not become obese throughout the study. Compared with LFD, HFD significantly reduced the food intake of mice during feeding (*p* < 0.05, Figure 1B).

### 3.3. Impact of W. coagulans and EA on Insulin Resistance in the High-Fat Diet-Fed Mice

Mice in the HFD group had the highest blood glucose levels 30 min after gavage of glucose (*p* < 0.05, Figure 2A), with a significant increase in the area under the curve (AUC), compared with the LFD group (*p* < 0.05, Figure 2B). The EA + BC2000 intervention treatment significantly reduced 30 min blood glucose levels, and it also reduced the AUC by 23% compared with the HFD group (*p* < 0.05). The synergistic effect of EA + BC2000 significantly reduced insulin levels compared with the HFD group (*p* < 0.05, Figure 2C) and thereby brought HOMA-IR to a near-normal level (Figure 2D). In particular, HOMA-IR was significantly lower in the HFD + EA + BC2000 group compared with HFD + EA (*p* < 0.05), but not in the HFD + EA + BC77 group. These indicate that the synergistic effect is superior to the effect of EA alone. Compared with BC77, BC2000 is more effective in maintaining glucose metabolic homeostasis and relieving insulin resistance.

### 3.4. Impact of W. coagulans and EA on Inflammatory Cytokines in Obese Mice

Compared with the LFD group, the plasma levels of LPS, hs-CRP, and Zonulin significantly increased in the HFD group (*p* < 0.05), as shown in Figure 3. In the HFD + EA group, the increase in plasma LPS and hs-CRP levels caused by HFD was significantly suppressed by EA (*p* < 0.05). Compared with the HFD + EA group, the addition of BC2000 reduced hs-CRP and Zonulin levels (*p* < 0.05), whereas the addition of BC77 only reduced hs-CRP levels (*p* < 0.05). Zonulin is an important intestinal permeability indicator. An elevated Zonulin level indicates increased intestinal permeability and an impaired mucosal barrier [17]. As a result, a large number of inflammatory cytokines (e.g., LPS, TNF-α, and IL-6) enter the blood circulation and liver tissue, causing low-grade systemic inflammation [18].

The impact of BC2000 on inflammatory cytokines in the plasma and liver of mice was also assessed. As shown in Figure 3D–F, HFD significantly increased the plasma IL-6 and TNF-α levels and the liver TNF-α level compared with LFD (*p* < 0.05). Compared with the HFD group, the plasma IL-6 level and the liver TNF-α level were decreased in the HFD + EA, HFD + EA + BC77, and HFD + EA + BC2000 groups (*p* < 0.05). The latter three groups showed no significant difference in the two levels (*p* > 0.05).

### 3.5. Impact of W. coagulans and EA on Cecum Microbiota

The microbiota in the mouse cecum was determined through 16S rDNA amplicon sequencing to further investigate the impact of BC2000 on the gut microbiota composition. Chao1 index is often used in ecology to indicate the total number of species (namely, the richness of gut microbiota). Higher Chao1 values indicate greater species richness. Shannon and Simpson indexes are used to estimate the diversity of microbiota. The alpha diversity of the cecum microbiota is shown in Figure 4. The Shannon index of the cecum microbiota in the HFD group was significantly lower than in the LFD group (*p* < 0.05), and the Simpson index was also reduced, but the decrease was not statistically significant (Figure 4A, B). This indicates that HFD decreased the diversity of cecum microbiota in obese mice. Compared with the HFD group, the Chao1 index of cecum microbiota was markedly reduced in the HFD + EA and HFD + EA + BC2000 groups (*p* < 0.05), but no significant difference was observed in the HFD + EA + BC77 group (Figure 4C). Beta diversity intuitively reflects the differences in microbial community composition among different samples. As shown in Figure 4D, the CPCoA based on the Bray–Curtis distance matrix showed significant differences in microflora between the LFD and HFD groups (ANOSIM, *R* = 0.86, *p* < 0.001). The differences suggest marked changes in the gut microbiota composition of HFD mice. Microbiota partially overlapped between the HFD + EA + BC2000 and LFD groups. The overlap indicates that BC2000 could ameliorate HFD-induced gut dysbiosis. Compared with the HFD + EA group, the dominant families of the cecum flora in the HFD + EA + BC2000 group especially contained *Planococcaceae, Bacillaceae, Micrococcaceae, Atocobiaceae, Eggerthellaceae*, and *Moraxelaceae* (Figure 4E). Among them, *Eggerthellaceae* can facilitate the conversion of EA to urolithin, which is thought to be responsible for the physiological effects of tannins and EA in foods [19]. Recent studies have shown that urolithin A prevents streptozotocin-induced diabetic cardiomyopathy in rats [20]. *Eggerthellaceae* abundance in HFD + EA + BC2000 was significantly higher than that in the HFD + EA group. This is mainly attributed to two species of the *Eggerthellaceae* family, *Adlercreutzia equolifaciens* and *Adlercreutzia mucosicola* (Figure 4F,G). In particular, they were significantly enriched in the HFD + EA + BC2000 group compared to HFD + EA + BC77 (*p* < 0.05).

Compared with LFD at the genus level (Table 2), HFD notably increased the relative abundance of *Limosilactobacillus, Lactobacillus,* and *Ileibacterium* in mice, while lowering that of *Akkermansia, Muribaculum, Alistipes,* and *Mucispirillum*. Compared with the HFD group, the HFD + EA + BC2000 group showed a significant increase in the relative abundance of *Muribaculum, Helicobacter, Sporobacter, Acetatifactor, Weizmannia* (*Weizmannia coagulans*) and *Alistipes* and an obvious decrease in that of *Sphingopyxis, Parasutterella, Gordonibacter, Psychrobacter, and Amedibacillus.* In the HFD + EA + BC2000 group, the relative content of *Limosilactobacillus* and *Ileibacterium* was higher due to BC2000 supplementation than that in the HFD + EA group.

In addition, the microbial functions were analyzed and compared among groups (Figure 5). Compared with the LFD group, the HFD group showed significant changes in 12 pathways. They included the super pathway of geranylgeranyldiphosphate, mevalonate pathway I, the super pathway of L-methionine biosynthesis (by sulfhydrylation), lipid IVA biosynthesis, and the super pathway of L-threonine metabolism. The relative abundance of five pathways in the HFD group was significantly higher than that in the LFD group, while that of the other seven pathways was markedly lower. These results indicate that HFD changed microbial functions in mice. The relative abundance of 13 pathways in the HFD + EA group and 10 pathways in the HFD + EA + BC77 group showed a notable decline compared with the HFD group, but a significant increase was observed in the relative abundance of 4 pathways in the HFD + EA + BC77 group. Compared with the HFD group, the HFD + EA + BC2000 group had an obvious decrease in the relative abundance of 11 pathways. The 11 pathways were fatty acid salvage, catechol degradation metadata-cleavage pathway), NAD biosynthesis II (from tryptophan), catechol degradation to 2-oxopent-4-enoate II, L-tryptophan degradation to 2-amino-3-carboxymuconate, L-tryptophan degradation XII (Geobacillus), L-tryptophan degradation IX, 2-amino-3-carboxymuconate semialdehyde degradation, 2-nitrobenzoate degradation I, 2-aminophenol degradation, and isopropanol biosynthesis.

### 3.6. Impact of BC2000 on Gene Expression in Mouse Liver

To investigate how BC2000 + EA affects the whole gene expression in HFD-induced obese mice, we further performed transcriptome sequencing analysis of mouse liver. As shown in Figure 6A, the CPCoA results showed that the gene expression profiles of the livers of mice in the HFD + EA + BC2000 group were significantly separated from the HFD group, the HFD + EA group, and the HFD + EA + BC77 group. Log2FC > 1.5 and *p* < 0.05 were used as cut-off criteria for the DEGs. As shown in Figure 6B, 424 genes were significantly upregulated and 268 genes were significantly downregulated in the HFD group compared with the LFD group. The HFD+EA group had 338 genes significantly upregulated and 348 genes significantly downregulated compared with the HFD group. The HFD+EA+BC2000 group had 435 genes significantly upregulated and 268 genes significantly downregulated compared to the HFD group. There were 290 genes significantly upregulated and 268 genes significantly downregulated in the HFD+EA+BC2000 group compared with the HFD+EA group.

GO enrichment analysis describes the biological processes and molecular functions of DEGs enrichment in each group of mice. Compared with the HFD group, the EA + BC2000 intervention activated biological processes such as PPAR signaling, fatty acid degradation and valine, leucine and isoleucine degradation, as shown in Figure 6C. Compared with EA alone, the addition of BC2000-activated retinol metabolism, cholesterol metabolism, and primary bile acid biosynthesis (Figure 6C). Figure 6D describes the molecular function of DEGs enrichment. The HFD + EA + BC2000 group activated autophagy compared with the HFD group. Next, interaction network analysis based on the pathway of KEGG revealed that the addition of EA + BC2000 significantly affected metabolic pathways related to obesity and immune response, such as autophagy, mitophagy, NOD-like receptor signaling pathway, mTOR signaling pathway, ErbB signaling pathway, insulin resistance, and other metabolic pathways (Figure 6E).

### 3.7. Insulin Signaling Gene Expression

To investigate the possible mechanisms by which the combined consumption of *W. coagulans* and EA can potentially drive insulin sensitivity, the insulin-signaling gene (peroxisome proliferator-activated receptor gamma, PPARG) and the expression of insulin pathway genes have been investigated. Compared with the HFD group, PPARG expression was only significantly upregulated in the HFD+EA+BC2000 group (*p* < 0.05), as shown in Table 3. Compared with the LFD group, the HFD resulted in a downregulation of PIK3CA, PIK3CB, PIK3R1, AKT1, and AKT3 expression, but the administration of EA+BC77 or EA+BC2000 reversed the reduction except PIK3CA. In addition, the addition of EA+BC2000 upregulated the expression of mTOR (*p* < 0.05).

### 3.8. Differential Gene KEGG Enrichment Analysis

As shown in Figure 7, the fatty acid degradation pathway and valine, leucine, and isoleucine degradation pathway showed that EA + BC2000 intervention activated the fatty acid degradation pathway and promoted the expression of enoyl-CoA hydratase, 3-hydroxyacyl-CoA dehydrogenase, and acetyl-CoA acyltransferase, which encode proteins mainly involved in fatty acid oxidation. In addition, HFD + EA + BC2000 intervention activated valine, leucine, and isoleucine degradation and promoted the expression of 2-oxoisovalerate dehydrogenase E1 component-α subunit, enoyl-CoA hydratase, cetoacaetyl coenzyme A, 3-hydroxyacyl coenzyme A dehydrogenase, and aldehyde dehydrogenase, encoding enzymes that mainly promoted the catabolism of branched-chain amino acids.

### 3.9. Correlation Analysis

The correlation analysis between gut microbiota and biochemical indicators (Figure 8) indicated that the genus *Akkermansia* was significantly negatively correlated with the liver TNF-α level and plasma Zonulin levels (*p* < 0.05). The genus *Alloprevotella* was negatively correlated with LPS, insulin, HOMA-IR index, and plasma IL-6 level (*p* < 0.05). The genus *Lactobacillus* was positively correlated with LPS, insulin, HOMA-IR index, AUC, and plasma levels of IL-6 and Zonulin (*p* < 0.05). These indicate that the mixture of BC2000 and EA could ameliorate insulin resistance, glucose tolerance and homeostasis, and inflammatory response by regulating the composition of gut microbiota.

## 4. Discussion

In this study, the combined intervention of *W. coagulans* BC2000 and EA significantly reduced HOMA-IR compared with EA alone, while the combined intervention of *W. coagulans* BC77 and EA did not differ significantly from EA alone. This indicates that the EA + BC2000 intervention was more effective in improving HFD-induced insulin resistance. The mechanism is closely related to the activation of cellular autophagy and mitosis in the liver, enhancement of the intestinal barrier and remodeling of the intestinal microbiota. Significant cell, animal, and human data support the idea that urolithin A increases autophagy and mitophagy [21]. This implies that BC2000 might enhance the conversion of EA to urolithins in the intestine. BC2000 promotes the growth of the potential EA transformation family *Eggerthellaceae,* especially *Adlercreutzia equolifaciens and Adlercreutzia mucosicola* which may involve in EA conversion and need further verification.

Previous research found that fasting plasma glucose and insulin levels significantly increased in HFD mice, and the increase was characterized by insulin resistance [22]. In this study, supplementation with the BC2000 and EA mixture ameliorated glucose tolerance and reduced the level and sensitivity of insulin in HFD-induced obese mice. These findings were consistent with another study [23] in which EA plus BC2000 intervention significantly regulated autophagy and mitophagy pathways. Autophagy can play an important role in various physiological and pathological processes such as cell homeostasis, ageing, immunity, tumorigenes, and neurodegenerative diseases [24]. Studies have shown that urolithin A, a metabolite of EA in the gut, can prolong the lifespan of nematodes and improve muscle function in rodents by inducing autophagy [25]. A recent study found that urolithin A ameliorates obesity-induced metabolic cardiomyopathy in mice by activating autophagy [26]. Urolithin A alleviates diabetes and pancreatic damage by activating the AKT/mTOR pathway to regulate insulin signaling and autophagy to reduce oxidation, inflammation and apoptosis in the pancreas of type 2 diabetic mice [27]. This is consistent with the results of our study. Compared with HFD+EA, HFD+EA+BC2000 significantly modulated mTOR signaling and autophagy. Therefore, we speculate that BC2000 promotes the conversion of EA to urolithin A, and urolithin A regulates the process of autophagic clearance by regulating mTOR signaling.

The urolithin A-like activation of autophagy induced by EA plus BC2000 may be attributed to *Adlercreutzia equolifaciens* and *Adlercreutzia mucosicola* from the *Eggerthellaceae* family. *Adlercreutzia equolifaciens* is an equol-producing bacterium [28]. It was found that *Adlercreutzia* was predominant in individuals characterized by urolithin A production [29]. Whether it is also involved in urolithin production requires further validation. The *Eggerthellaceae* family could convert EA into urolithins [19]. This metabolite is much better absorbed than EA and is thought to contribute to the health effects of ellagitannin and EA [30]. Urolithin A and urolithin B can alleviate weight gain caused by high-fat diets by modulating the gut microbiota [31]. *Eggerthellaceae* were significantly elevated in the gut microbiota of mice on high-fat and high-sucrose diets that were intervened by berry fiber [32]. This is consistent with our findings. In a recent study, urolithin A and urolithin B were administered to Wister rats, and it was found that urolithin B could increase the relative abundance of *Alistipes* in the rat intestine [33]. *Alistipes* can produce short-chain fatty acids [34], which are closely related to insulin resistance [35]. In this study, *Alistipes* were significantly enriched in the HFD+EA+BC2000 group. These results suggest that BC2000 promotes the conversion of EA to urolithin by modulating the intestinal microbiota. In addition, Anhê et al. [36] described that the consumption of polyphenol-rich cranberry extract prevented obesity and insulin resistance in mice induced by a high-fat and high-sugar diet by modulating the intestinal microbiota, particularly the abundance of *Akkermansia*. In this study, a high-fat diet resulted in a significant reduction in *Akkermansia* abundance that was alleviated by the addition of EA+BC2000. Furthermore, we did not find that supplementation with BC77 activated the autophagy pathway in mouse liver. This indicates that the effect of *W. coagulans* is strain-specific. In this study, *Weizmannia*, the only genus in which *C. coagulans* exists, was significantly upregulated in the intestinal microbiota of mice supplemented with BC2000 only. This may indicate that BC2000 is more likely to colonize the gut than BC77. These results suggest that the addition of BC2000 to promote EA’s prevention of high-fat-diet-induced insulin resistance is closely related to the reshaping of the gut microbiota.

Research has shown that *W. coagulans* can inhibit the HFD-induced expression of inflammatory cytokines IL-1β and IL-6 in the liver to improve systemic inflammation and insulin resistance [37]. IL-6 and TNF-α can induce insulin resistance in the liver and adipocytes by reducing the phosphorylation of insulin receptor substrate (IRS) or by inhibiting IRS transcription [38]. In the present study, the combination of *W. coagulans* and EA in food significantly reduced the plasma levels of endotoxins LPS, hs-CRP, Zonulin, TNF-α, and IL-6 and relieved HFD-induced insulin resistance. In particular, the addition of BC2000 significantly reduced plasma Zonulin levels compared with EA consumption alone. Elevated plasma Zonulin levels indicate the increased permeability of the intestinal mucosa and thus damage to the mucosal barrier [18]. The possible reason is analyzed as follows: HFD damages the intestinal mucosa. Through the damaged mucosa, a large number of inflammatory factors enter the veins or lymphatic system. LPS binds to TLR4 on the cell surface to activate the nuclear transcription factor κ (NFκB). The activation promotes the expression of target genes IL-6 and TNF-α to cause low-grade systemic inflammation and insulin resistance [39]. However, *W. coagulans* prevents gut dysbiosis, thereby inhibiting excessive LPS generation and preventing elevated LPS levels in the blood, which can cause metabolic endotoxemia. This mechanism helps prevent chronic inflammation and insulin resistance. The expression levels of NF-κB and TLR4 will be determined in follow-up work to analyze how BC2000 combined with EA reduces inflammatory cytokines.

In addition to the aforementioned cellular autophagy and mTOR signaling pathways, combined consumption of BC2000 and EA significantly upregulated the expression of PPARG, a key transcription factor regulating the development and proper metabolism of adipocytes. PPARG may have implications for the regulation of other genes important to the insulin pathway [40,41]. Moreover, PPARG regulates the expression of adiponectin—responsible for increasing the sensitivity of cells to insulin [40,41]. Many studies have confirmed that PPARG expression is reduced in insulin resistance, in both cell and human studies [42,43]. An earlier study showed that EA-rich tea extracts prevented impaired insulin sensitivity in the body by activating the expression of PPARG [44]. This is consistent with the findings of this paper. In addition, BC2000+EA can regulate the insulin-signaling pathway (P13K/AKT/mTOR). Insulin binds to the α subunits of the insulin receptor and activates the tyrosine kinase in the β subunit. Once tyrosine kinase of the β subunit is activated, it promotes the threonine phosphorylation of IRS-1 and in turn stimulates the activation of the PI3K/AKT pathway. AKT directly promotes glucose uptake and transmits signals to downstream mTOR. Meanwhile, mTOR plays an extremely important role in cellular autophagy and metabolic processes. A recent study found that urolithin A intervened in type 2 diabetic mice and found that the pancreatic protective effect of urolithin A in diabetes was mediated by its regulation of autophagy and AKT/mTOR signaling pathways, specifically urolithin A, which increased the phosphorylation levels of p-AKT and p-mTOR [27]. In the HFD + EA + BC2000 group, PI3K and mTOR were significantly upregulated. This reinforces our hypothesis that BC2000 promotes the conversion of EA to urolithin A, which regulates autophagy clearance by modulating P13K/AKT/mTOR signaling, thereby slowing down the prevention of high-fat-induced insulin resistance in mice. 

Furthermore, BC2000 and EA significantly regulated liver pathways of fatty acid metabolism, cholesterol metabolism, bile acid metabolism, retinol metabolism and PPAR signaling. Activation of the retinol metabolic pathway by BC2000 and EA may also be related to insulin resistance inhibition by retinoic acid. Retinoic acid can improve the symptoms of diabetes in mice by promoting insulin secretion [45]. Retinoic acid upregulates glucokinase activity and expression [46], which not only promotes secretion but also produces energy that can be supplied to proinsulin synthesis [47].

The combined consumption of BC2000 and EA significantly upregulated these enzymes involved in fatty acid metabolism and cholesterol metabolism. This indicates that the addition of BC2000 helps fatty acid degradation to provide energy for the body. Elevated levels of fatty acids in the blood are a major factor driving insulin secretion [48]. Bile acids are the major products of cholesterol metabolism in the liver. This study showed that HFD+EA+BC2000 intervention activated the cholesterol metabolism pathway and bile acid metabolism pathway, indicating that HFD+EA+BC2000 could promote the conversion of cholesterol into bile and steroid hormones, thereby regulating metabolism.

## 5. Conclusions

In conclusion, BC2000 increased gut *Eggerthellaceae* with EA-transforming potential and induced the activation of the hepatic autophagy pathway, thereby assisting EA in playing a role in the prevention of insulin resistance due to high-fat diets. BC2000 can be added to EA-rich foods as a probiotic.

## Figures and Tables

**Figure 1 nutrients-14-04206-f001:**
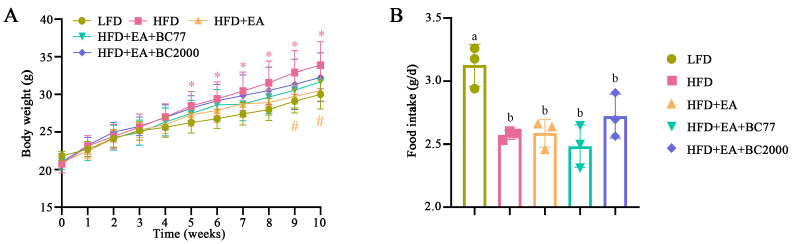
Impact of BC2000 on body weight (**A**) and food intake (**B**) of HFD mice. * indicates a significant difference compared with the LFD group (*p* < 0.05); # indicates a significant difference compared with the HFD group (*p* < 0.05); Different lowercase letters above bars indicate significant differences among groups (*p* < 0.05). n = 10/group. Values show mean ± SD. Tukey’s test based on a one-way analysis of variance was used for multiple comparisons. LFD: low-fat-diet group; HFD: high-fat-diet group; HFD+EA: high-fat-diet supplemented with ellagic acid; HFD+EA+BC77: high-fat-diet supplemented with ellagic acid and *W. coagulans* BC77; HFD+EA+BC2000: high-fat-diet supplemented with ellagic acid and *W. coagulans* BC2000.

**Figure 2 nutrients-14-04206-f002:**
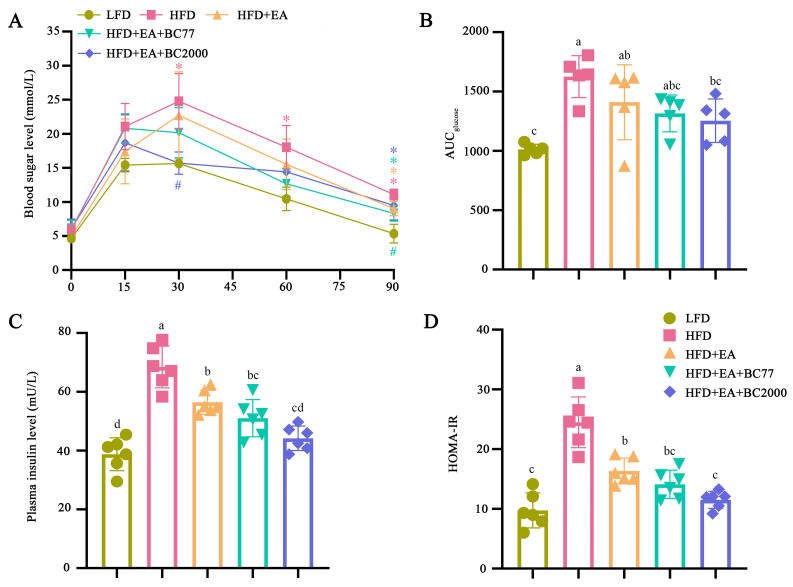
Impact of BC2000 on glucose homeostasis and insulin resistance in obese mice: (**A**) blood sugar level; (**B**) AUC; (**C**) plasma insulin level; (**D**) HOMA-IR. * indicates a significant difference compared with the LFD group (*p* < 0.05). # indicates a significant difference compared with the HFD group (*p* < 0.05). Different lowercase letters above bars indicate significant differences among groups (*p* < 0.05). Blood sugar level and AUC (n = 5/group). Plasma insulin level and HOMA-IR (n = 6/group). Values show mean ± SD. Tukey’s test based on a one-way analysis of variance was used for multiple comparisons. LFD: low-fat-diet group; HFD: high-fat-diet group; HFD + EA: high-fat-diet supplemented with ellagic acid; HFD + EA + BC77: high-fat-diet supplemented with ellagic acid and *W. coagulans* BC77; HFD + EA + BC2000: high-fat-diet supplemented with ellagic acid and *W. coagulans* BC2000.

**Figure 3 nutrients-14-04206-f003:**
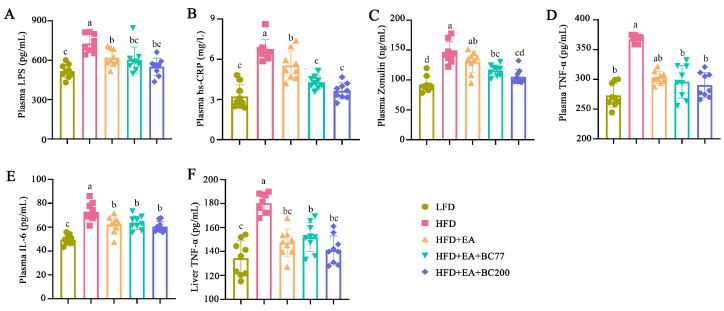
Impact of BC2000 on inflammatory cytokines in obese mice: (**A**) plasma LPS; (**B**) plasma hs-CRP; (**C**) plasma Zonulin; (**D**) plasma TNF-α; (**E**) plasma IL-6; (**F**) liver TNF-α. Different letters indicate significant differences (*p* < 0.05). n = 9/group. Values show mean ± SD. When variances of the study results were homogeneous, Tukey’s test based on one-way analysis of variance (ANOVA) was used for multiple comparisons. When the variances were not homogeneous, the Kruskal–Wallis test was used. LFD: low-fat-diet group; HFD: high-fat-diet group; HFD+EA: high-fat-diet supplemented with ellagic acid; HFD + EA + BC77: high-fat-diet supplemented with ellagic acid and *W. coagulans* BC77; HFD + EA + BC2000: high-fat-diet supplemented with ellagic acid and *W. coagulans* BC2000.

**Figure 4 nutrients-14-04206-f004:**
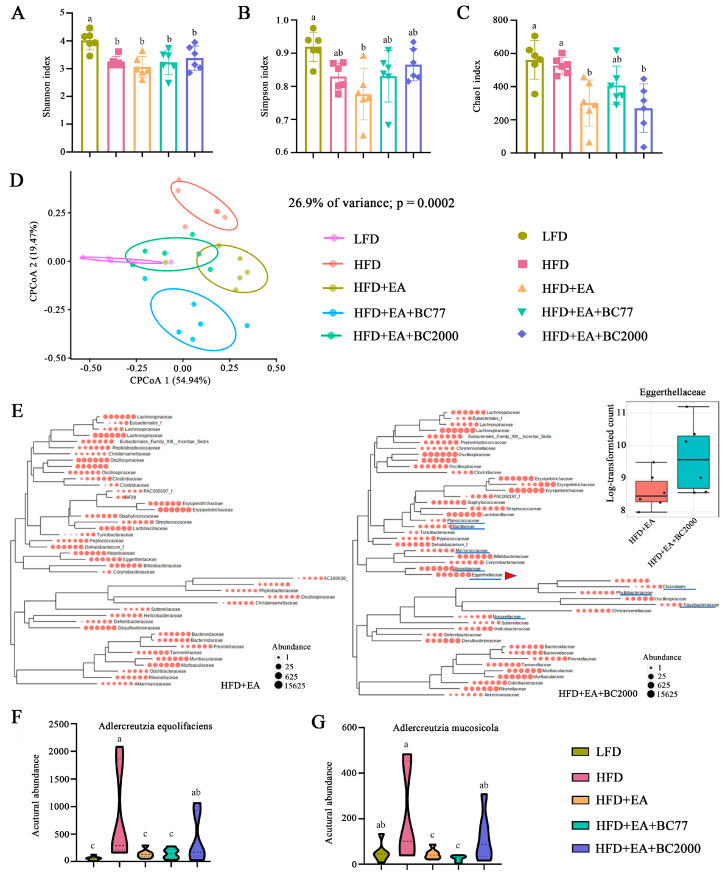
Impact of BC2000 on cecum microbiota in obese mice: (**A**) Shannon index; (**B**) Simpson index; (**C**) Chao1 index; (**D**) CPCoA; (**E**) phylogenetic trees of gut microbiota families of HFD+EA and HFD+EA+BC2000 groups and relative content of Eggerthellaceae (inset). (**F**) The abundance of *Adlercreutzia equolifaciens* of *Eggerthellaceae*. (**G**) The abundance of *Adlercreutzia mucosicola* of *Eggerthellaceae*. Different letters indicate significant differences (*p* < 0.05). n = 6/group. Values show mean ± SD. When variances in the study results were homogeneous, Tukey’s test based on one-way analysis of variance (ANOVA) was used for multiple comparisons. When the variances were not homogeneous, the Kruskal–Wallis test was used. LFD: low-fat-diet group; HFD: high-fat-diet group; HFD + EA: high-fat-diet supplemented with ellagic acid; HFD + EA + BC77: high-fat-diet supplemented with ellagic acid and *W. coagulans* BC77; HFD + EA + BC2000: high-fat-diet supplemented with ellagic acid and *W. coagulans* BC2000.

**Figure 5 nutrients-14-04206-f005:**
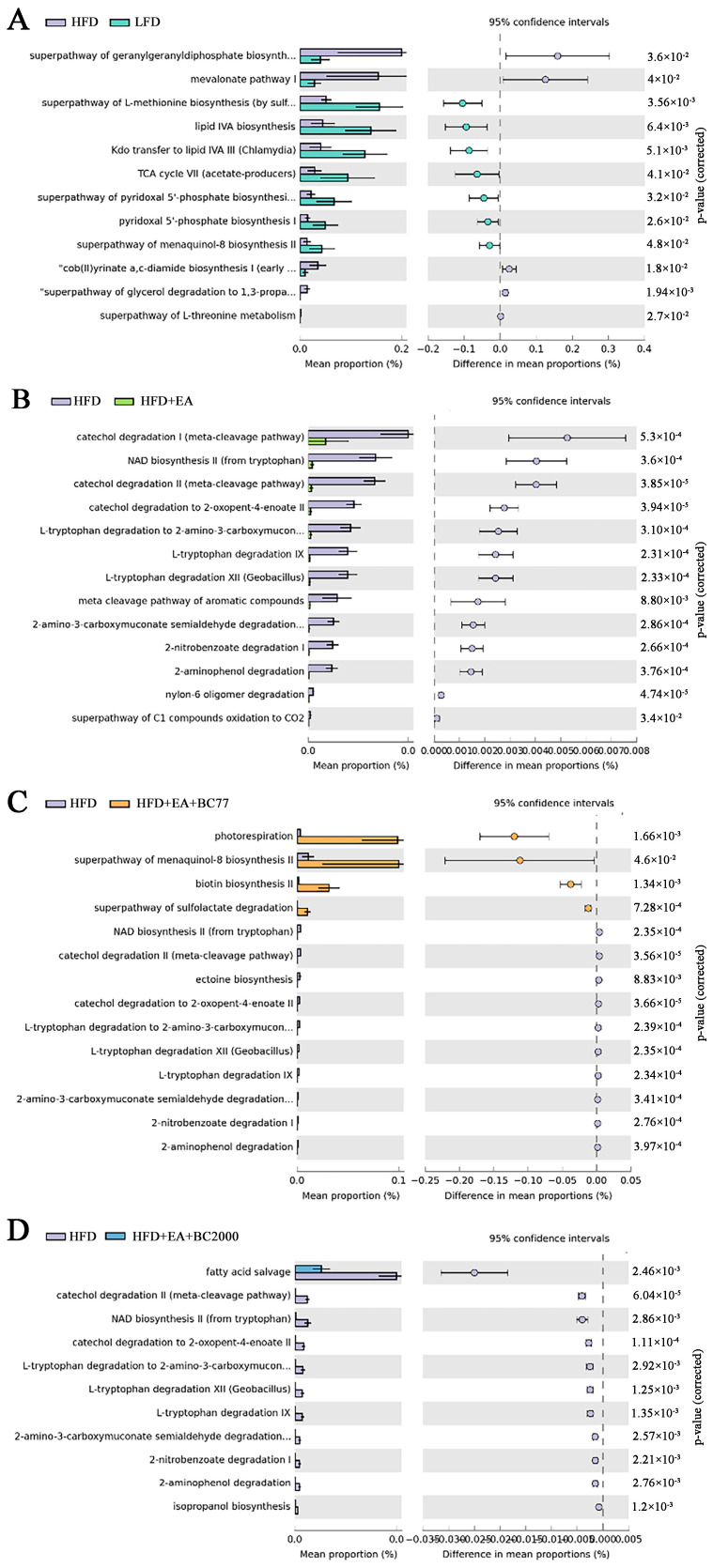
Significant differences (FDR-adjusted *p*-value < 0.05) between groups in MetaCyc Gene Pathways predicted by PICRUSt2. (**A**) HFD VS LFD; (**B**) HFD VS HFD + EA; (**C**) HFD VS HFD + EA + BC77; (**D**) HFD VS HFD + EA + BC2000. n = 6/group. Data are expressed with log2FC. Differential expression is analyzed using the edgeR software. LFD: low-fat-diet group; HFD: high-fat-diet group; HFD + EA: high-fat-diet supplemented with ellagic acid; HFD + EA + BC77: high-fat-diet supplemented with ellagic acid and *W. coagulans* BC77; HFD + EA + BC2000: high-fat-diet supplemented with ellagic acid and *W. coagulans* BC2000.

**Figure 6 nutrients-14-04206-f006:**
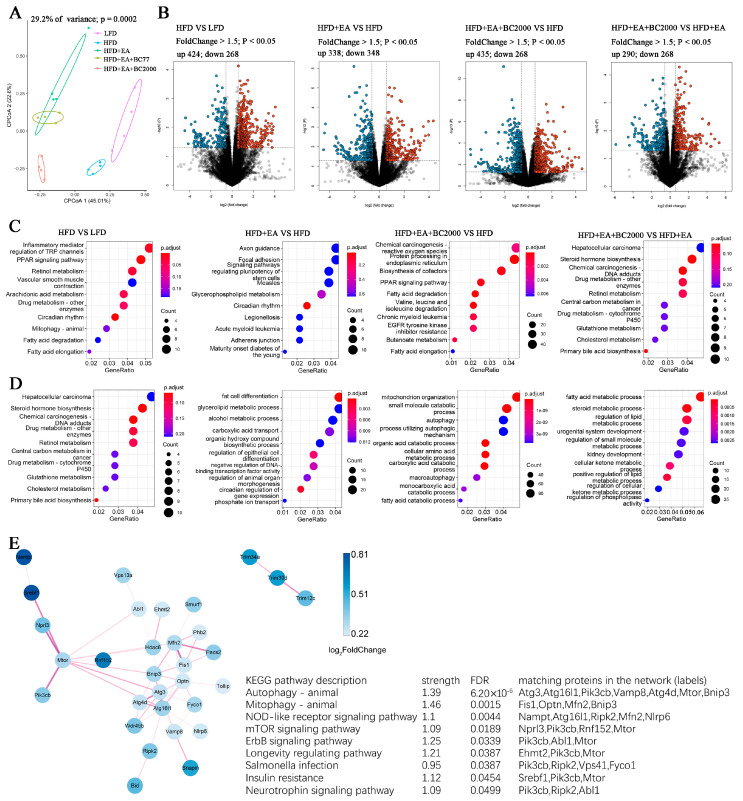
Impact of BC2000 on gene expression in mouse liver; (**A**) CPCoA; (**B**) volcano plot; (**C**) GO enrichment analysis of biological processes; (**D**) GO enrichment analysis of molecular functions; (**E**) interaction network map of the KEGG pathway based on differentially expressed genes. n = 4/group. LFD: low-fat-diet group; HFD: high-fat-diet group; HFD + EA: high-fat-diet supplemented with ellagic acid; HFD + EA + BC77: high-fat-diet supplemented with ellagic acid and *W. coagulans* BC77; HFD + EA + BC2000: high-fat-diet supplemented with ellagic acid and *W. coagulans* BC2000.

**Figure 7 nutrients-14-04206-f007:**
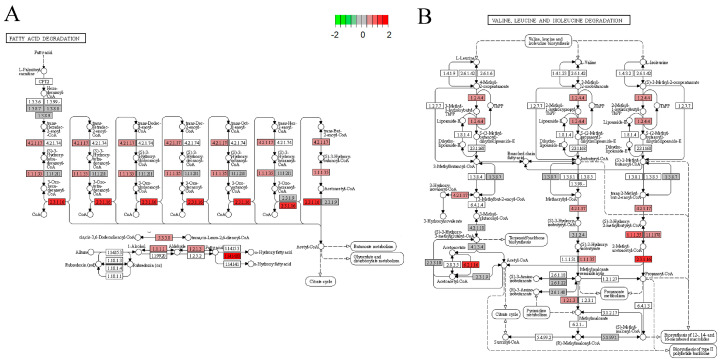
KEGG pathway analysis. (**A**) KEGG pathway map of fatty acid degradation signaling pathway; (**B**) KEGG pathway map of valine, leucine and isoleucine degradation signaling pathway. n = 4/group. LFD: low-fat-diet group; HFD: high-fat-diet group; HFD + EA: high-fat-diet supplemented with ellagic acid; HFD + EA + BC77: high-fat-diet supplemented with ellagic acid and *W. coagulans* BC77; HFD + EA + BC2000: high-fat-diet supplemented with ellagic acid and *W. coagulans* BC2000.

**Figure 8 nutrients-14-04206-f008:**
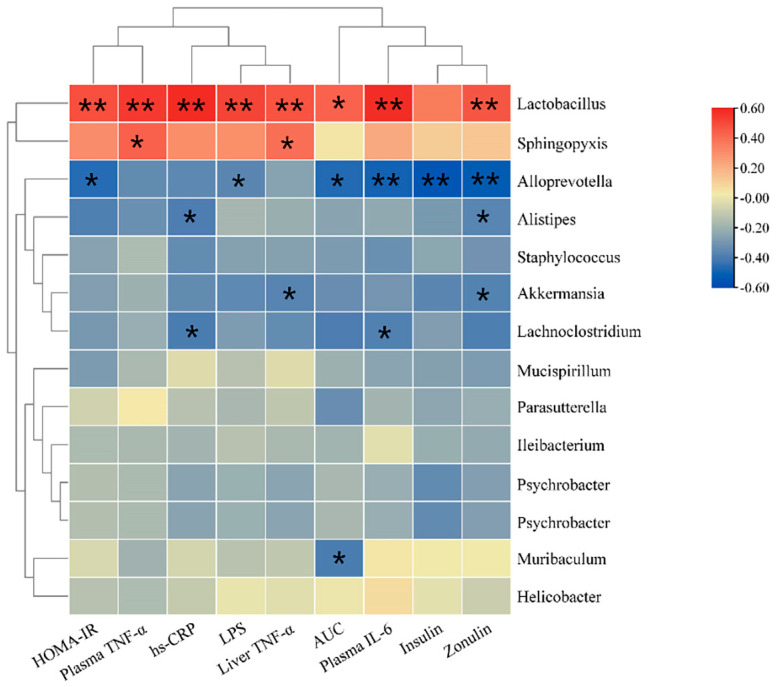
Heat map for correlations between biochemical indicators and gut microbiota. The X-axis and Y-axis of the heat map represent biochemical indicators and gut microbiota, respectively. The *R* and *p* values are calculated. The *R* values are expressed with different colors. The color card on the right side shows the color ranges of different *R* values. The “*” symbol indicates a significant correlation at the 0.05 level; ** indicates s significant correlation at the 0.01 level. The top clusters represent the clustering of biochemical indicators, and the left-side clusters represent the clustering of gut microbiota.

**Table 1 nutrients-14-04206-t001:** Composition of animal feed (g/kg).

Ingredients	LFD	HFD	HFD+EA	HFD+EA+BC77	HFD+EA+BC2000
Casein	189.6	231	231	231	231
L-Cystine	2.8	3	3	3	3
Corn Starch	479.8	83.9	83.6	83.5	83.5
Maltodextrin	118.5	115	115	115	115
Sucrose	65.2	199.3	199.3	199.3	199.3
Cellulose	47.4	58	58	58	58
Soybean oil	23.7	29	29	29	29
Lard	19	204.7	204.7	204.7	204.7
Mineral Mix	9.5	12	12	12	12
Dicalcium Phosphate	12.3	15	15	15	15
Calcium Carbonate	5.2	6.3	6.3	6.3	6.3
Potassium Citrate·H_2_O	15.6	19	19	19	19
Vitamin Mix	9.5	12	12	12	12
Choline Bitartrate	1.9	2	2	2	2
Cholesterol	0	9.8	9.8	9.8	9.8
Ellagic acid	0	0	0.3	0.3	0.3
Probiotic powder	0	0	0	0.1	0.1
Total (g)	1000	1000	1000	1000	1000
Calories (kcal/g)					
Protein	20%	20%	20%	20%	20%
Carbohydrate	70%	35%	35%	35%	35%
Fat	10%	45%	45%	45%	45%
Total	100	100	100	100	100

Note: LFD: low-fat-diet group; HFD: high-fat-diet group; HFD+EA: high-fat-diet supplemented with ellagic acid; HFD+EA+BC77: high-fat-diet supplemented with ellagic acid and *W. coagulans* BC77; HFD+EA+BC2000: high-fat-diet supplemented with ellagic acid and *W. coagulans* BC2000.

**Table 2 nutrients-14-04206-t002:** Mouse cecum bacteria significantly affected by BC2000.

ID	HFD vs. LFD		HFD+EA vs. HFD		HFD+EA+BC77 vs. HFD		HFD+EA+BC2000 vs. HFD	
	log2FC	*p*	log2FC	*p*	log2FC	*p*	log2FC	*p*
*Alloprevotella*	2.87	<0.01	–2.57	<0.01	–2.18	0.02	–2.55	0.01
*Muribaculum*	–1.42	0.03	1.40	0.04	2.13	<0.01	2.28	<0.01
*Limosilactobacillus*	8.08	<0.01	–2.43	<0.01	–3.55	<0.01		
*Alistipes*	–2.24	<0.01			2.50	<0.01	3.86	<0.01
*Akkermansia*	–5.89	<0.01						
*Lactobacillus*	6.14	<0.01			–2.00	<0.01		
*Sphingopyxis*			–5.05	<0.01	–3.93	<0.01	–5.46	<0.01
*Ileibacterium*	3.48	<0.01	–4.09	<0.01			4.20	<0.01
*Helicobacter*	–2.21	0.02	3.68	<0.01	2.39	0.03	5.03	<0.01
*Sporobacter*			2.86	<0.01	1.88	0.02	1.92	0.04
*Parasutterella*			–2.70	<0.01	–2.57	<0.01	–2.71	<0.01
*Acetatifactor*			3.93	<0.01	4.63	<0.01	4.18	<0.01
*Mucispirillum*	–2.57	0.02	2.72	<0.01	3.35	<0.01		
*Lachnoclostridium*			1.84	0.01	1.77	0.01	2.29	<0.01
*Weizmannia*					2.70	<0.01	5.47	<0.01
*Gordonibacter*	3.88	<0.01					–3.28	<0.01
*Psychrobacter*			–3.22	0.01	–3.78	<0.01	–2.46	0.03
*Bifidobacterium*	2.60	<0.01	–2.05	<0.01	–2.67	<0.01		
*Frisingicoccus*	2.34	<0.01	–1.51	0.03			–2.62	<0.01
*Anaerosporobacter*	2.54	<0.01			–2.15	<0.01	–1.86	0.01
*Turicibacter*	–3.19	<0.01	2.98	0.01				
*Sporobacter*			2.86	<0.01	1.88	0.02	1.92	0.04
*Parasutterella*			–2.70	<0.01	–2.57	<0.01	–2.71	<0.01
*Acetatifactor*			3.93	<0.01	4.63	<0.01	4.18	<0.01
*Amedibacillus*			–2.68	0.01			–2.13	0.04

Note: Log2FC means Log2 fold change. *p* < 0.05 is considered statistically significant. The genus-level differences of microbiota were analyzed by the edgeR-based Wald test for pairwise comparison between groups. n = 6/group. LFD: low-fat-diet group; HFD: high-fat-diet group; HFD + EA: high-fat-diet supplemented with ellagic acid; HFD + EA + BC77: high-fat-diet supplemented with ellagic acid and *W. coagulans* BC77; HFD + EA + BC2000: high-fat-diet supplemented with ellagic acid and *W. coagulans* BC2000.

**Table 3 nutrients-14-04206-t003:** Insulin-signaling gene expression.

ID	HFD vs. LFD		HFD+EA vs. HFD		HFD+EA+BC77 vs. HFD		HFD+EA+BC2000 vs. HFD	
	log2FC	*p*	log2FC	*p*	log2FC	*p*	log2FC	*p*
PPARG	0.17	0.73	–0.27	0.33	–0.08	0.76	0.82	0.00
PIK3CA	–0.02	0.90	–0.14	0.47	–0.26	0.07	–0.37	0.00
PIK3CB	–0.24	0.20	0.30	0.14	0.23	0.11	0.39	0.01
PIK3R1	–0.08	0.70	0.24	0.30	0.59	0.00	0.64	0.00
AKT1	–0.08	0.65	0.10	0.61	0.02	0.90	0.02	0.86
AKT3	–0.10	0.74	–0.07	0.79	0.12	0.58	0.06	0.78
mTOR	0.11	0.54	0.34	0.07	0.33	0.02	0.29	0.03

Note: Log2FC means Log2 fold change. *p* < 0.05 is considered statistically significant. The differences in genes were analyzed by the edgeR-based Wald test for pairwise comparison between groups. n = 4/group. LFD: low-fat-diet group; HFD: high-fat-diet group; HFD + EA: high-fat-diet supplemented with ellagic acid; HFD + EA + BC77: high-fat-diet supplemented with ellagic acid and *W. coagulans* BC77; HFD + EA + BC2000: high-fat-diet supplemented with ellagic acid and *W. coagulans* BC2000.

## Data Availability

The raw data of 16S rRNA gene libraries generated during this study are publicly available at the Sequence Read Archive (SRA) portal of NCBI under accession number PRJNA856975.

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
