# Peer review of "Weizmannia coagulans BC2000 Plus Ellagic Acid Inhibits High-Fat-Induced Insulin Resistance by Remodeling the Gut Microbiota and Activating the Hepatic Autophagy Pathway in Mice"

_nutrients, 2022, doi:10.3390/nu14194206_

Round 1
Reviewer 1 Report
Jin et al. describe that food supplementation of a specific microbiota Weizmannia coaglulans together with ellagic acid could improve high-fat diet-induced insulin resistance in wildtype C57 mice. Although a lot of work had been done to support their conclusion, the constant inconsistency of method description and results analysis largely raises the concern of credibility of this study:
1, In the Methods section, there is an inconsistency in administrating Weizmannia coaglulans. In text line 85, the authors indicate “dose at 10^8 CFU/animal/day while in Table 1, the authors suggested that probiotic powder was supplemented together with food at 0.1 g/kg. How exactly the microbiota is administrated and how much was dosed?
2, In the Methods section, there is an inconsistency in how the glucose tolerance test is conducted. In line 99, the authors indicated oral glucose administration while in line 100 the authors said that they injected the glucose.
3, In the Results, line 188, the authors describe "each HFD-fed group was smaller". However, according to Figure 2B, the HFD significantly increases the AUC.
4, In the Results section, it is very confusing which specific tissue was used for RNA sequencing as in the title and figure legend, it is indicated as liver while in the text line 312, the authors use ‘mouse ileum’.
5, None of the authors was affiliated with an institute at Qingdao University while in the method section they claim the animal studies were conducted in Qingdao University with an approved protocol number.
Other than that, there are other issues that need to be solved before considering resubmitting or publishing.
1, The rationale for using the Weizmannia coaglulans, instead of the other specific microbiota, needs to be further highlighted. For example, the authors could potentially make a comparison of the capability of inducing anaerobic conditions within the commercially available microbiota that are permitted by the food regulation rules.
2, In the Results section, the authors use HOMA-IR as the only indicator of insulin resistance. It is suggested to also include other surrogate markers, for example, the phosphorylation levels of Akt signaling.
3. Please include individual points of measuring parameters together with mean and SD in the bar chart.
4, In the Methods section, line 139, the authors only use 4 samples out of 10 total samples per group to conduct the RNA extraction experiments without specifying the reasons. It is suggested to increase the number and explain why.
5, In the Results, line 186, the HFD+EA group did not significantly alter the glycemia excursion within the HFD group (Figure 2A) while the authors suggest that ‘was significantly reversed by in the HFD+EA’.
Author Response
Response to Reviewer 1 Comments
Dear Reviewers,
Thank you very much for your valuable comments. We have made changes carefully to the text according to the comments and suggestions.
以下是我们为回应审稿人的意见而对稿件所做的所有更改的详细信息。所有更改在修订后的手稿中都以黄色突出显示。
评论
Jin等人描述说,食物补充特定的微生物群魏茨曼尼亚粘胶聚糖与鞣花酸可以改善野生型C57小鼠的高脂肪饮食诱导的胰岛素抵抗。尽管已经做了很多工作来支持他们的结论,但方法描述和结果分析的持续不一致在很大程度上引起了本研究可信度的担忧:
回应:非常感谢您的宝贵意见,我们仔细检查了全文并进行了充分的修改。这些更改在修订后的稿件中以黄色突出显示。
第1点.在“方法”部分中,在管理粘液酸魏茨曼尼亚方面存在不一致。在文本第85行中,作者指出“剂量为10 ^ 8 CFU /动物/天,而在表1中,作者建议益生菌粉与食物一起补充0.1g / kg。微生物群究竟是如何管理的,剂量是多少?
答复1:已修订(L92-95)。
饲料在1000克饲料中含有0.1克冻干BC2000或BC77粉末。益生菌粉含量为4×10 ^ 11 CFU / g。基于小鼠的采食量为2.3-3.3克/天,BC2000或BC77摄入量为9.2×10^7-1.32×10^8 CFU/天/动物。
第2点.在方法部分,葡萄糖耐量测试的进行方式不一致。在第99行中,作者表示口服葡萄糖给药,而在第100行中,作者说他们注射了葡萄糖。
答复2:已修订(L113-116)。
小鼠在夜间无水禁食后用400mg / mL葡萄糖(每公斤体重5mL)进行加氢。在强饲术后分别在0,15,30,60和90分钟从小鼠尾静脉收集血液。收集的血液用于确定小鼠的血糖水平。
第3点.在结果第188行中,作者描述了“每个HFD喂养的组都较小”。然而,根据图2B,高频抗坏血病显著增加了AUC。
答复3:已修订(L212-216)。
HFD组的小鼠在葡萄糖灌注后30分钟具有最高的血糖水平(P<0.05,图2A),与LFD组相比,曲线下的面积(AUC)显着增加(P<0.05,图2B)。EA + BC2000干预治疗显着降低了30分钟的血糖水平,并且与HFD组相比,它还将AUC降低了23%(P <0.05)。
第4点.在结果部分,非常令人困惑的是,在标题和图例中,哪个特定组织用于RNA测序,它被指示为肝脏,而在文本行312中,作者使用“小鼠回肠”。
响应 4:已修订 (L351)
肝脏组织用于RNA测序。
第5点.没有一个作者隶属于青岛大学的研究所,而在方法部分,他们声称动物研究是在青岛大学进行的,具有批准的方案编号。
答复5:第二作者,党洪阳,隶属于新疆大学。曾任青岛大学营养健康研究所联合培养。动物实验由邓鸿阳在青岛大学营养与健康研究所进行人工培养。因此,动物伦理学批准号来自青岛大学。作者信息已修订并添加 (L10)
其他作者信息: 5 青岛大学营养与健康研究所, 青岛, 中国
除此之外,在考虑重新提交或发布之前,还有其他问题需要解决。
第1点.需要进一步强调使用粘胶体魏茨曼尼亚而不是其他特定微生物群的基本原理。例如,作者可以潜在地比较食品监管规则允许的市售微生物群中诱导厌氧条件的能力。
Response 1: Previous tests by our group compared the probiotic properties such as heat, acid and bile salt tolerance of Weizmannia coaglulans with five commercially available Weizmannia coaglulans from different countries. We found that Weizmannia coaglulans BC2000 showed good resistance to heat, acid and bile salts. The relevant data are described in the Revised Manuscript (L76-79, L178-191 and Supplementary Material).
Because nuts are rich in ellagic acid, traditional probiotics such as Bifidobacterium and Lactobacillus are temperature sensitive (Tripathi and Giri 2014). They do not survive long at room temperature. Therefore, they cannot be added to nut foods that are kept at room temperature. However, Weizmannia coaglulans' good resistance to heat, acid and bile salts can solve this problem. This reason was added to the Introduction section of the Revised Manuscript (L52-56).
Point 2. In the Results section, the authors use HOMA-IR as the only indicator of insulin resistance. It is suggested to also include other surrogate markers, for example, the phosphorylation levels of Akt signaling.
Response 2: Unfortunately, it was not possible to complete the Western Blot test for Akt phosphorylation levels because our animal samples were almost exhausted. To more fully justify that EA+BC2000 could alleviate insulin resistance, we supplemented the expression of insulin signalling genes (Peroxisome proliferator-activated receptor gamma, PPARG) and insulin pathway genes (PIK3-AKT-mTOR) obtained by transcriptome sequencing (Result: L383-392, Table 3 and Discussion: L513-536).
Point 3. Please include individual points of measuring parameters together with mean and SD in the bar chart.
Response 3: All bar charts were revised.
Point 4. In the Methods section, line 139, the authors only use 4 samples out of 10 total samples per group to conduct the RNA extraction experiments without specifying the reasons. It is suggested to increase the number and explain why.
Response 4: We commissioned Biotech Biotechnology Co., Ltd to perform a mouse liver transcriptomic analysis. Transcriptome sequencing analysis is expensive. We had limited funds for the experiment, so four samples were selected for sequencing. The number of samples was the same as in other studies where transcriptomics was performed (Wei et al. 2021)(Stazi et al. 2021)(Zhao et al. 2022).
Point 5. In the Results, line 186, the HFD+EA group did not significantly alter the glycemia excursion within the HFD group (Figure 2A) while the authors suggest that ‘was significantly reversed by in the HFD+EA’.
Response 5: Revised (L212-216).
Mice in the HFD group had the highest blood glucose levels 30 min after gavage of glucose (P < 0.05, Figure 2A), with a significant increase in the area under the curve (AUC), compared to the LFD group (P < 0.05, Figure 2B). The EA + BC2000 intervention treatment significantly reduced 30 min blood glucose levels and it also reduced the AUC by 23% compared to the HFD group (P < 0.05).

Reviewer 2 Report
This study to investigate Weizmannia coagulans BC2000 plus ellagic acid inhibits high-fat-induced insulin resistance by remodeling gut microbiota and activating hepatic autophagy pathway in mice. The author aims of this study was to investigate the synergistic effects and mechanisms of supplementation with the probiotic Weizmannia coagulans (W. coagulans) on the prevention of insulin resistance by Ellagic acid.
Figure 1 shows that the food intake of the high-fat diet is 10% lower than that of the normal diet, and the weight gain is significantly increased by more than 10%fu at the end. What is the reason for the significant difference of 15%? Is it because of feeding effects or individual differences?
Figure 1 shows that the food intake of the high-fat diet is 10% lower than that of the normal diet, and the weight gain is significantly increased by more than 10%fu at the end. What is the reason for the significant difference of 15%? Is it because of feeding effects or individual differences?
Figure 2 shows that FD + EA + BC2000 groups, but no significant difference was observed in the HFD + EA + BC77 group. There was no difference between HOMA-IR and AUG. Do you consider other statistical analysis methods?
Figure 3 shows that FD + EA + BC2000 groups but no significant difference was observed in the HFD + EA + BC77 group but the article title defines that bc2000 has an improvement effect. How to explain it?
Figure 4 shows that microbiota was observed in the FD + EA + BC2000 groups a little difference was observed in the HFD + EA + BC77 group The authors should indicate whether these differences are key factors in the effect
The authors analysed gene expression in the mouse liver but many literatures point out that metabolism is not the only genetic influence in this tissue. Could there be more explanation for the inhibition of high-fat-induced insulin resistance?
Conclusions indicate that BC2000 increases intestinal Eggerthellaceae with EA-transforming potential and induces activation of the hepatic autophagy pathway, thereby assisting EA to play a role in preventing insulin resistance. BC2000 can be added to EA rich foods as a probiotic Is there more information or data to confirm that activation of hepatic autophagy is relevant in the prevention of insulin resistance. Overall, this article is well done and worthy of publication in this journal. This study to investigate Weizmannia coagulans BC2000 plus ellagic acid inhibits high-fat-induced insulin resistance by remodeling gut microbiota and activating hepatic autophagy pathway in mice. The author aims of this study was to investigate the synergistic effects and mechanisms of supplementation with the probiotic Weizmannia coagulans (W. coagulans) on the prevention of insulin resistance by Ellagic acid.
Figure 1 shows that the food intake of the high-fat diet is 10% lower than that of the normal diet, and the weight gain is significantly increased by more than 10%fu at the end. What is the reason for the significant difference of 15%? Is it because of feeding effects or individual differences?
Figure 1 shows that the food intake of the high-fat diet is 10% lower than that of the normal diet, and the weight gain is significantly increased by more than 10%fu at the end. What is the reason for the significant difference of 15%? Is it because of feeding effects or individual differences?
Figure 2 shows that FD + EA + BC2000 groups, but no significant difference was observed in the HFD + EA + BC77 group. There was no difference between HOMA-IR and AUG. Do you consider other statistical analysis methods?
Figure 3 shows that FD + EA + BC2000 groups but no significant difference was observed in the HFD + EA + BC77 group but the article title defines that bc2000 has an improvement effect. How to explain it?
Figure 4 shows that microbiota was observed in the FD + EA + BC2000 groups a little difference was observed in the HFD + EA + BC77 group The authors should indicate whether these differences are key factors in the effect
The authors analysed gene expression in the mouse liver but many literatures point out that metabolism is not the only genetic influence in this tissue. Could there be more explanation for the inhibition of high-fat-induced insulin resistance?
Conclusions indicate that BC2000 increases intestinal Eggerthellaceae with EA-transforming potential and induces activation of the hepatic autophagy pathway, thereby assisting EA to play a role in preventing insulin resistance. BC2000 can be added to EA rich foods as a probiotic Is there more information or data to confirm that activation of hepatic autophagy is relevant in the prevention of insulin resistance
The discussion section should address which exercise has a clear improvement statement against which probiotic.
What is the reason for systematically sorting out the differences and improvements in the probiotic?
The description of the conclusion is slightly inappropriate, whether it is beneficial to exercise or the improvement of probiotic supplementation by ordinary people.
Author Response
Dear Reviewers,
Thank you very much for your valuable comments. We have made changes carefully to the text according to the comments and suggestions.
Here are the details of all the changes we have made to the manuscript in response to reviewers’ comments. All the changes are highlighted in yellow in the revised manuscript.
Comments
This study investigates Weizmannia coagulans BC2000 plus ellagic acid inhibits high-fat-induced insulin resistance by remodelling gut microbiota and activating hepatic autophagy pathway in mice. The author aims of this study was to investigate the synergistic effects and mechanisms of supplementation with the probiotic Weizmannia coagulans (W. coagulans) on the prevention of insulin resistance by Ellagic acid.
Point 1. Figure 1 shows that the food intake of the high-fat diet is 10% lower than that of the normal diet, and the weight gain is significantly increased by more than 10%fu at the end. What is the reason for the significant difference of 15%? Is it because of feeding effects or individual differences?
Response 1: Compared to low-fat feeds, high-fat feeds contain a lot of lard, and the equivalent amount of high-fat feeds and low-fat feeds, high-fat feeds contain more energy. 100 g of lard contains approximately 900 kcal of calories. Lard has a high fat content and the fat in lard is a saturated acid that is not easily absorbed by the body, leading to a build-up of fat in the body. It is easy to lead to obesity in the long term.
Point 2. Figure 1 shows that the food intake of the high-fat diet is 10% lower than that of the normal diet, and the weight gain is significantly increased by more than 10%fu at the end. What is the reason for the significant difference of 15%? Is it because of feeding effects or individual differences?
Response 2: Same as point 1.
Point 3. Figure 2 shows the HFD + EA + BC2000 groups, but no significant difference was observed in the HFD + EA + BC77 group. There was no difference between HOMA-IR and AUC. Do you consider other statistical analysis methods?
Response 3: The statistical analysis methods of the Revised Manuscript have been revised. When variances of the study results were homogeneous, Tukey’s test based on one-way analysis of variance (ANOVA) was used for multiple comparisons. When the variances were not homogeneous, the Kruskal-Wallis test. was used. P < 0.05 indicates a statistically significant difference. The statistical analysis in the Revised Manuscript methods has been revised (L172-175). The relevant statistical methods are described in the figure notes and table notes.
The Description in Figure 2 has been revised (L212-222).
Mice in the HFD group had the highest blood glucose levels 30 min after gavage of glucose (P < 0.05, Figure 2A), with a significant increase in the area under the curve (AUC), compared to the LFD group (P < 0.05, Figure 2B). The EA + BC2000 intervention treatment significantly reduced 30 min blood glucose levels and it also reduced the AUC by 23% compared to the HFD group (P < 0.05). The synergistic effect of EA + BC2000 significantly reduced insulin levels compared to the HFD group (P < 0.05, Figure 2C) and thereby brought HOMA-IR to a near normal level (Figure 2D). In particular, HOMA-IR was significantly lower in the HFD + EA + BC2000 group compared to HFD + EA (P < 0.05), but not in the HFD + EA + BC77 group. These indicate that the synergistic effect is superior to the effect of EA alone. Compared to BC77, BC2000 is more effective in maintaining glucose metabolic homeostasis and relieving insulin resistance.
Point 4. Figure 3 shows that FD + EA + BC2000 groups but no significant difference was observed in the HFD + EA + BC77 group but the article title defines that bc2000 has an improvement effect. How to explain it?
Response 4: Figure 3 shows the expression levels of inflammatory cytokines in plasma and liver, which can only represent the effect of the production of insulin resistance in mice on this aspect of organismal inflammation. The title of the article is a synthesis of all the findings of this study.
As shown in Figure 2, plasma insulin levels as well as HOMA-IR were significantly lower in the HFD+EA+BC200 group compared to the HFD + EA group, with no significant difference in the HFD + EA + BC77 group. The AUC was significantly lower in the HFD + EA + BC200 group compared to the HFD group, but there was no significant difference between the HFD and HFD + EA + BC77 groups. This indicates that BC2000 can better promote the physiological activity of EA. BC2000 is more effective in maintaining glucose metabolic homeostasis and relieving insulin resistance.
As shown in Figure 4, the Eggerthellaceae family can facilitate the conversion of EA to urolithins in the intestine, and urolithins are more readily absorbed by the body than ellagic acid (Selma et al. 2017). A large number of studies have shown that urolithins can regulate glycolipid metabolism (Albasher, Alkahtani, and Al-Harbi 2022) (Huang et al. 2022) (Tuohetaerbaike et al. 2020). Two of the species of the Eggerthellaceae family: Adlercreutzia equolifaciens and Adlercreutzia mucosicola were significantly enriched in the HFD + EA + BC2000 group compared to the HFD + EA + BC77 group.
As shown in Figure 6D, we also demonstrated that HFD + EA + BC2000 can significantly affect the autophagy pathway compared to HFD + EA. The intestinal metabolite urolithin of EA can prevent insulin resistance by regulating the autophagy pathway, autophagy is closely related to insulin (Huang et al. 2022) (Ávalos et al. 2022) (Tuohetaerbaike et al. 2020).
Based on these results, we conclude that the combination of BC2000 and EA can better prevent insulin resistance in mice on a high-fat diet.
Point 5. Figure 4 shows that microbiota was observed in the FD + EA + BC2000 groups a little difference was observed in the HFD + EA + BC77 group. The authors should indicate whether these differences are key factors in the effect
Response 5: Revised (L281-291).
Compared with the HFD + EA group, the dominant families of the cecum flora in the HFD + EA + BC2000 group especially contained Planococcaceae, Bacillaceae, Micrococcaceae, Atocobiaceae, Eggerthellaceae and Moraxelaceae (Figure 4E). Among them, Eggerthellaceae can facilitate the conversion of EA to urolithin, which is thought to be responsible for the physiological effects of tannins and EA in foods (Selma et al. 2017). Recent studies have shown that urolithin A prevents streptozotocin-induced diabetic cardiomyopathy in rats (Albasher, Alkahtani, and Al-Harbi 2022). Eggerthellaceae abundance in HFD + EA + BC2000 was significantly higher than that in the HFD + EA group. This is mainly attributed to two species of the Eggerthellaceae family, Adlercreutzia equolifaciens and Adlercreutzia mucosicola (Figure 4F, G). In particular, they were significantly enriched in the HFD + EA + BC2000 group compared to HFD + EA + BC77 (P < 0.05).
Point 6. The authors analysed gene expression in the mouse liver but many literatures point out that metabolism is not the only genetic influence in this tissue. Could there be more explanation for the inhibition of high-fat-induced insulin resistance?
Response 6: Revised.
In the Discussion section of the Revised Manuscript, we discuss the effects of alterations in the microbiota of the mouse cecum (L466-493) and the expression of plasma inflammatory cytokines (L494-512) on insulin resistance. The effects of combined consumption of BC2000 and ellagic acid on high-fat diet-induced insulin resistance in mice were investigated from three aspects: intestinal, plasma and liver.
Point 7. Conclusions indicate that BC2000 increases intestinal Eggerthellaceae with EA-transforming potential and induces activation of the hepatic autophagy pathway, thereby assisting EA to play a role in preventing insulin resistance. BC2000 can be added to EA-rich foods as a probiotic. Is there more information or data to confirm that activation of hepatic autophagy is relevant in the prevention of insulin resistance. Overall, this article is well done and worthy of publication in this journal. This study investigates Weizmannia coagulans BC2000 plus ellagic acid and inhibits high-fat-induced insulin resistance by remodelling gut microbiota and activating hepatic autophagy pathway in mice. The author aims of this study was to investigate the synergistic effects and mechanisms of supplementation with the probiotic Weizmannia coagulans (W. coagulans) on the prevention of insulin resistance by Ellagic acid.
Response 7: We complemented the expression of insulin signalling genes (Peroxisome proliferator-activated receptor gamma, PPARG) and insulin pathway genes (PIK3-AKT-mTOR) obtained by transcriptome sequencing (Result: L383-392, Table 3 and Discussion: L513-536). A recent study showed that UroA alleviates diabetes and pancreatic damage by activating the AKT/mTOR pathway to regulate insulin signalling and autophagy to reduce oxidation, inflammation and apoptosis in the pancreas of type 2 diabetic mice (Tuohetaerbaike et al. 2020). This is consistent with our findings. Thus, suggesting that BC2000 may regulate the autophagic clearance process through PIK3-AKT-mTOR by promoting the conversion of EA to urolithin A. Many studies have shown that autophagy can alleviate insulin resistance (Du et al. 2022) (Wang et al. 2022) (Zhang et al. 2021). In the Discussion Section of the Revised Manuscript, we revised (L457-465 and L527-536).
Point 8. Figure 1 shows that the food intake of the high-fat diet is 10% lower than that of the normal diet, and the weight gain is significantly increased by more than 10%fu at the end. What is the reason for the significant difference of 15%? Is it because of feeding effects or individual differences?
Response 8. Same as point 1.
Point 9. Figure 1 shows that the food intake of the high-fat diet is 10% lower than that of the normal diet, and the weight gain is significantly increased by more than 10%fu at the end. What is the reason for the significant difference of 15%? Is it because of feeding effects or individual differences?
Response 9. Same as point 1.
Point 10. Figure 2 shows that FD + EA + BC2000 groups, but no significant difference was observed in the HFD + EA + BC77 group. There was no difference between HOMA-IR and AUG. Do you consider other statistical analysis methods?
Response 10. Same as point 3.
Point 11. Figure 3 shows that FD + EA + BC2000 groups but no significant difference was observed in the HFD + EA + BC77 group but the article title defines that bc2000 has an improvement effect. How to explain it?
Response 11. Same as point 4.
Point 12. Figure 4 shows that microbiota was observed in the FD + EA + BC2000 groups a little difference was observed in the HFD + EA + BC77 group The authors should indicate whether these differences are key factors in the effect
Response 12. Same as point 5.
Point 13: The authors analysed gene expression in the mouse liver, but many literatures point out that metabolism is not the only genetic influence in this tissue. Could there be more explanation for the inhibition of high-fat-induced insulin resistance?
Response 13. Same as point 6.
Point 14: Conclusions indicate that BC2000 increases intestinal Eggerthellaceae with EA-transforming potential and induces activation of the hepatic autophagy pathway, thereby assisting EA to play a role in preventing insulin resistance. BC2000 can be added to EA-rich foods as a probiotic Is there more information or data to confirm that activation of hepatic autophagy is relevant in the prevention of insulin resistance
Response 14: Same as point 7.
Point 15. The discussion section should address which exercise has a clear improvement statement against which probiotic.
Response 15: Revised (L437-443)
In this study, the combined intervention of W. coagulans BC2000 and EA significantly reduced HOMA-IR compared to EA alone, while the combined intervention of W. coagulans BC77 and EA did not differ significantly compared to EA alone. This suggests that the EA + BC2000 intervention was more effective in improving HFD-induced insulin resistance. The mechanism is closely related to the activation of cellular autophagy and mitosis in the liver, enhancement of the intestinal barrier and remodeling of the intestinal microbiota.
Point 16. What is the reason for systematically sorting out the differences and improvements in the probiotic?
Response 16: Revised (L437-447)
In this study, the combined intervention of W. coagulans BC2000 and EA significantly reduced HOMA-IR compared to EA alone, while the combined intervention of W. coagulans BC77 and EA did not differ significantly compared to EA alone. This suggests that the EA + BC2000 intervention was more effective in improving HFD-induced insulin resistance. The mechanism is closely related to the activation of cellular autophagy and mitosis in the liver, enhancement of the intestinal barrier and remodeling of the intestinal microbiota. Significant cell, animal, and human data support the idea that urolithin A increases autophagy and mitophagy (Andreux et al. 2019). This implies that BC2000 might enhance the conversion of EA to urolithins in the intestine. BC2000 promotes the growth of the potential EA transformation family Eggerthellaceae, especially Adlercreutzia equolifaciens and Adlercreutzia mucosicola which may involve in EA conversion and need further verification.
Point 17. The description of the conclusion is slightly inappropriate, whether it is beneficial to exercise or the improvement of probiotic supplementation by ordinary people.
Response 17: Revised (L553-556)
In conclusion, BC2000 increased gut Eggerthellaceae with EA transforming potential and induced the activation of the hepatic autophagy pathway, thereby assisting EA in playing a role in the prevention of insulin resistance due to high-fat diets. BC2000 can be added to EA-rich foods as a probiotic.

Round 2
Reviewer 1 Report
I thank the authors for the modification of basically every comment I provided. The whole manuscript is extensively improved. I have no further comments and believe this paper is ready for publication.